# Acoustic Simulations Applied to the Garden of Rufolo's Villa in Ravello: Comparison between Different Scenarios

Antonella Bevilacqua [1,*], Giovanni Amadasi [2], Gino Iannace [3] and Amelia Trematerra [3]

1   Department of Architecture and Engineering, University of Parma, 43100 Parma, Italy
2   SCS-ControlSys_Vibro-Acoustic, 35011 Padova, Italy; g.m.amadasi@scs-controlsys.com
3   Department of Architecture, University of Campania L. Vanvitelli, 81031 Aversa, Italy;
    gino.iannace@unicampania.it (G.I.); amelia.trematerra@unicampania.it (A.T.)
*   Correspondence: antonella.bevilacqua@unipr.it

**Abstract:** This manuscript treats the acoustic analysis of a garden located in Rufolo's villa, south of Italy, which has already been studied to install some acoustic panels to improve the response across the seating area. After a campaign of acoustic measurements, acoustic simulations have been conducted based on three specific scenarios, highlighting the effectiveness of a new acoustic shell (project C) over the existing conditions (project A) and option A shell (project B). The results coming from the simulations show that the values of the main acoustic parameters are significantly improved with the installation of a scientifically designed acoustic shell, to be closer to the optimal ranges. The comparison among different projects highlights that the acoustic response of the existing conditions is not optimal, and that a marginal improvement was found with the addition of vertical panels mounted at the perimeter of the stage in combination with suspended ceiling above the stage. This research study is addressed to potentially auralize the voice or sound of famous artists that would be performing within the new acoustic shell; furthermore, the outputs could even implement the website of the Rufolo's villa for marketing purposes.

**Keywords:** architectural acoustics; Villa Rufolo; Ravello Festival; acoustic measurements; acoustic simulations; acoustic shell





## 1. Introduction

Ravello is a small village belonging to the Amalfi coast and is located 350 m above sea level. This village is the center of the Ravello Festival, an international event that has been taking place for about 50 years. During this event that runs between June and September, many musical events are organized every year, including music (both opera and symphonic) concerts, spectacles based on prose, classic ballets, movies, meetings, and conferences [1]. The events take place in different buildings of Ravello, such as St John's church, St Mary's church of Gradillo, the main square of the town, and inside the garden of Rufolo's villa, which is very fascinating due to its wide panorama and spectacular belvedere over the sea. Many concerns regarding outdoor spaces are related to the lack of acoustics due to the absence of useful reflecting surfaces, typical of ancient and/or unroofed theatres [2]. From these case studies, numerous acoustic measurements and simulations have been carried out in order to improve the existing conditions and design reflecting panels to be installed on the stage or around the seating areas [3]. It is important that the acoustic simulations are run with the presence of an audience and in unoccupied conditions to understand the amount of absorption that varies with different percentage of occupancy [4].

Generally, the presence of acoustic screens around the perimeter of the stage improves the acoustics only marginally, localized in the proximity of the performing area, but not for far distances [5]. This condition is identified in this article as option A shell (project B), which is proposed by the managers of the Ravello Festival. This is differentiated from

the existing condition (project A) and from another acoustic shell proposed by the authors (project C).

The reflecting panels should be designed in such a way to address the sound energy towards the audience and allow the values of the main acoustic parameters to be within the optimal range [6]. This is the key factor that influenced the design of the option B shell (project C) with the side walls designed like petals while the ceiling is characterized by reflecting panels with a convex profile.

It should be remembered that the shows taking place where the acoustics have not been properly investigated are likely to be characterized by failure due to the evident weaknesses. For the garden of Rufolo's villa, the option A shell (project B) is not acoustically satisfying since the panels 3 m high installed at the perimeters of the stage are not very effective; from these outcomes, it is recommended to improve the acoustics by creating new favorable reflections [7].

Before any simulations, a campaign of acoustic measurements was undertaken in accordance with ISO 3382 [8], which represents the baseline to carry out the evaluation of the acoustic effects of the three specific projects: the existing conditions (project A), the option A shell (project B) as suggested by the managers of the Ravello Festival, and a new shell proposed by the authors based on the acoustic investigations (project C). The validation of this study was performed digitally with the design of virtual models baked in Ramsete software 3.13 [9], which can be implemented, while the option B shell (project C) will be built in reality and be acoustically tested.

This article introduces a brief history of the garden of Rufolo's villa in combination with the development of the Ravello Festival. A dedicated chapter treats the characteristics of Wagner's garden, where the acoustic measurements have been carried out with different scenarios. After the comments on the measured results, the paper introduces the two options of an acoustic shell: the one proposed by the managers of the organization, and another one proposed by the authors and designed on the basis of the scientific outcomes.

## 2. Historical Background of Rufolo's Villa

The Rufolo's villa was built around the 13th century, instructed by the Rufolo's family to demonstrate the importance of this family in town. Built in Moresco style, the Rufolo's villa fell into disuse for different centuries until it was completely abandoned. The value of this residential property was discovered in 19th century by the rich Scottish botanist, Sir Francis Neville Reid, who designed the garden of a terrace that is located on the top of a cliff [1], as shown in Figure 1. In May of 1880, the composer Richard Wagner visited Ravello becoming inspired by the garden of Rufolo's villa to create the scenery of the second act of the Parsifal. In 1930, year of the Wagner's death, a great music concert was organized in the garden of the Rufolo's villa, an event that was decided to be regularly repeated from 1953 onwards, with the name of the Ravello Festival. At the beginning, the ensemble was located in the garden and afterwards, the location of the musicians was established on a stage, built externally to the garden, suspended at 13 m height, between the sky and the sea.

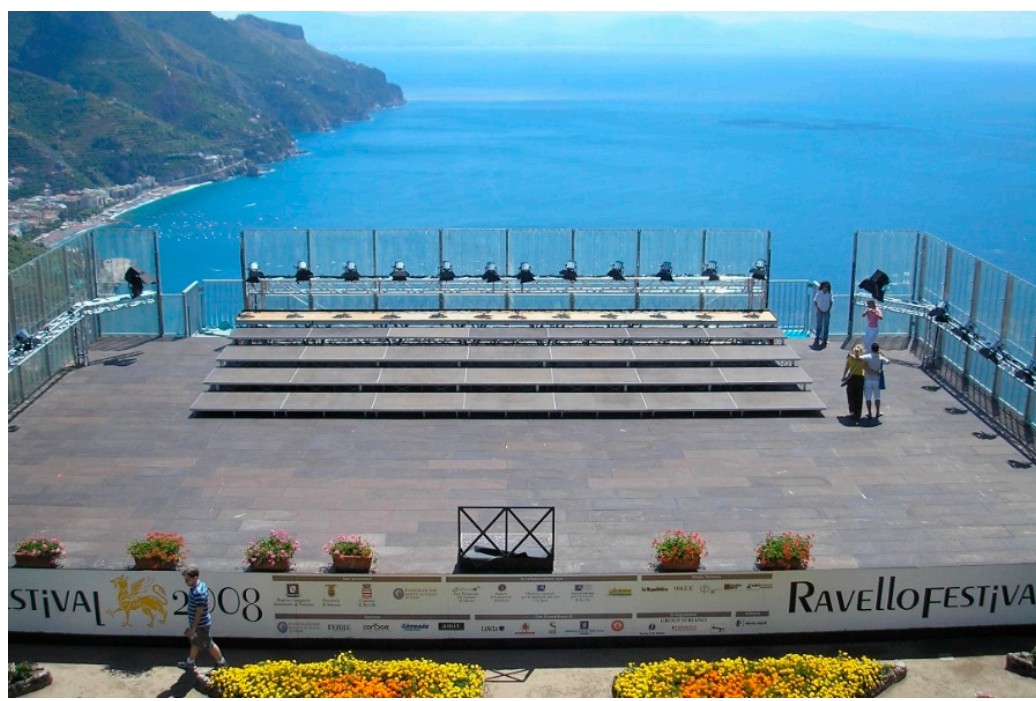

**Figure 1.** View of the panorama from the terrace of Rufolo's villa, in Ravello, Campania, Italy.

### 3. The Garden of Rufolo's Villa Where Richard Wagner Took Inspiration

A suspended stage is built in the garden of Rufolo's villa for all the time of the festival. The stage floor is suspended at 13 m from the level of the garden by a steel-frame structure. The stage is 23 m wide and 16 m long. The audience is located in a tribune, also composed of a steel-frame structure, that has a capacity of up to 800 spectators. The shortest distance between the stage and the tribune is about 7 m. At the perimeter of the stage, there are transparent acrylic panels, that are 1.5 m high and remain for symphonic executions. When opera is to be performed, the external panels mounted at the perimeter of the stage are about 3.5 m high, with the purpose of improving the acoustic quality of the music, as is the managers' belief.

### 4. Acoustic Measurements according to ISO 3382-1

Acoustic measurements have been conducted in the garden of Rufolo's villa. The equipment consists of an omnidirectional sound source (B&K OmniPower 4292-L, Bruel & Kajer, Darmstadt, Germany) placed in two positions on to the stage, and fourteen positions across the tribunes taken by the omnidirectional microphone (B&K 4155, ½-inch), following the criteria explained in ISO 3382 [8]. The sound source was fed with an MLS (Maximum Length Sequence) signal generated by a personal computer and amplified by a power amplifier. The sound source was placed at 1.6 m height from the stage floor, while the microphone was at 1.2 m from the tribune floor, simulating the height of the ears related to a sitting listener.

Figure 2 illustrates the positions where the equipment was installed during the measurements. The survey was performed on September 2008, during daytime, with an external temperature around 23–25 °C and an average wind speed < 3 m/s. The outdoor variance was considered during the survey [10]. The measurements were carried out with the following conditions:

- without any panels, and
- with the presence of transparent acrylic panels, floor-mounted at the three-sides perimeter of the stage, that were 3.5 m high.

  These panels were temporarily installed until a design of a new shell is in place.

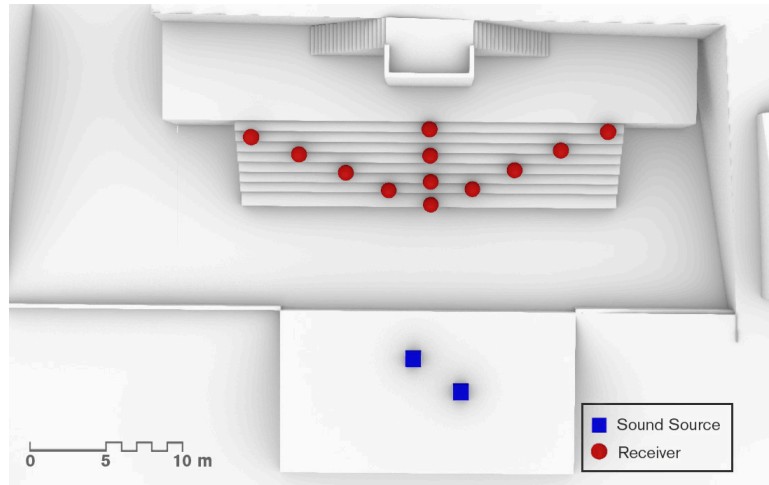

**Figure 2.** Equipment position during the acoustic measurements.

## 5. Comments upon Measured Results

The main acoustic parameters taken into consideration for the analysis of this particular space were the early decay time (EDT), reverberation time (T20), music clarity (C80), and definition (D50). The considered spectrum ranged between 125 Hz and 4 kHz. The results summarized in Figure 3 were averaged for all the source and receiver positions. The comparison includes the acoustic response measured in two specific conditions, without and with transparent acrylic panels that were 3.5 m high.

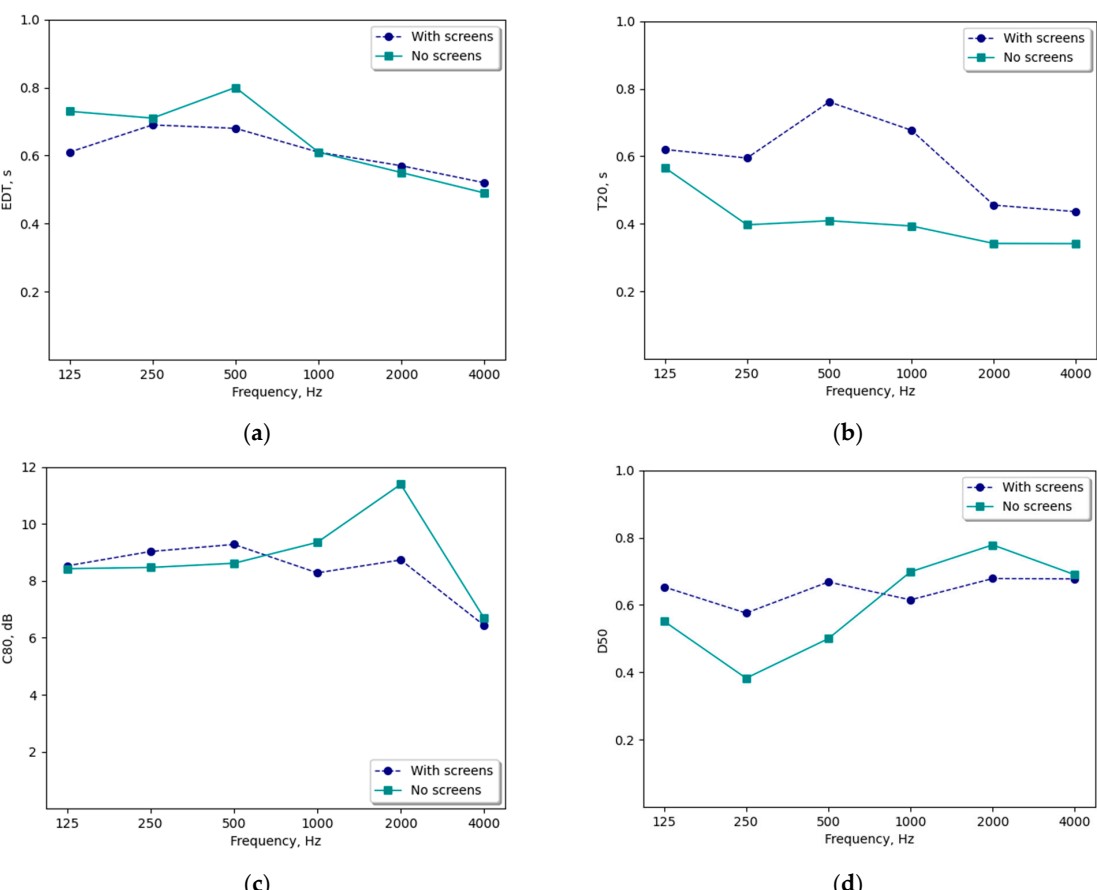

**Figure 3.** Average measured results related to the garden of Rufolo's villa: EDT (**a**), T20 (**b**), C80 (**c**), and D50 (**d**).

Figure 3a shows that a negligible difference was revealed between the two scenarios in relation to the EDT results. The EDT results, with and without screens, fluctuated around 0.7 s at low frequencies with a slight downward trend up to 4 kHz, where the values were around 0.5 s. Overall, the screens did not contribute to a significant acoustic improvement for the early reflections, which should be increased to be close to 1.8 s, as per the criteria set for enclosed spaces [11].

Figure 3b shows that the T20 results without screens fluctuate around 0.4 s, which is low for good acoustics of musical performances. The addition of 3.5 m screens improved marginally the reverberation, reaching 0.6 s for all the octaves and 0.8 s at 500 Hz, only.

The clarity index, with and without screens, fluctuated around 8 dB, with an upward peak equal to 11.5 dB at 2 kHz in the absence of any screens, as shown in Figure 3c. These C80 values were highly above the upper threshold set to +2 dB, meaning that the music (both vocal and instrumental) is clearer than what should be [12].

Figure 3d shows that the definition is around 0.4 and 0.6 at low frequencies and around 0.6 and 0.7 at high frequencies, without and with screens, respectively. These D50 values are considered suitable for musical performances [13]. The presence of screens flattened the response across the bandwidth to be more uniformly distributed without peaks at any octaves [14].

## 6. Design of New Acoustic Shells

### 6.1. Option A Shell: Proposal from Managers

One of the acoustic shells that was used temporarily in the garden of the Rofolo's villa was proposed by the managers of the Ravello Festival. It consisted of the floor-mounted installation of acrylic panels around the three-sided perimeter of the stage. These panels were 3.5 m high, waved to scatter the sound, and avoid standing waves between parallel surfaces. In addition, some suspended wooden panels had the same inclination towards the audience, installed at the same height from the stage floor. Figure 4 shows the views of this option A shell.

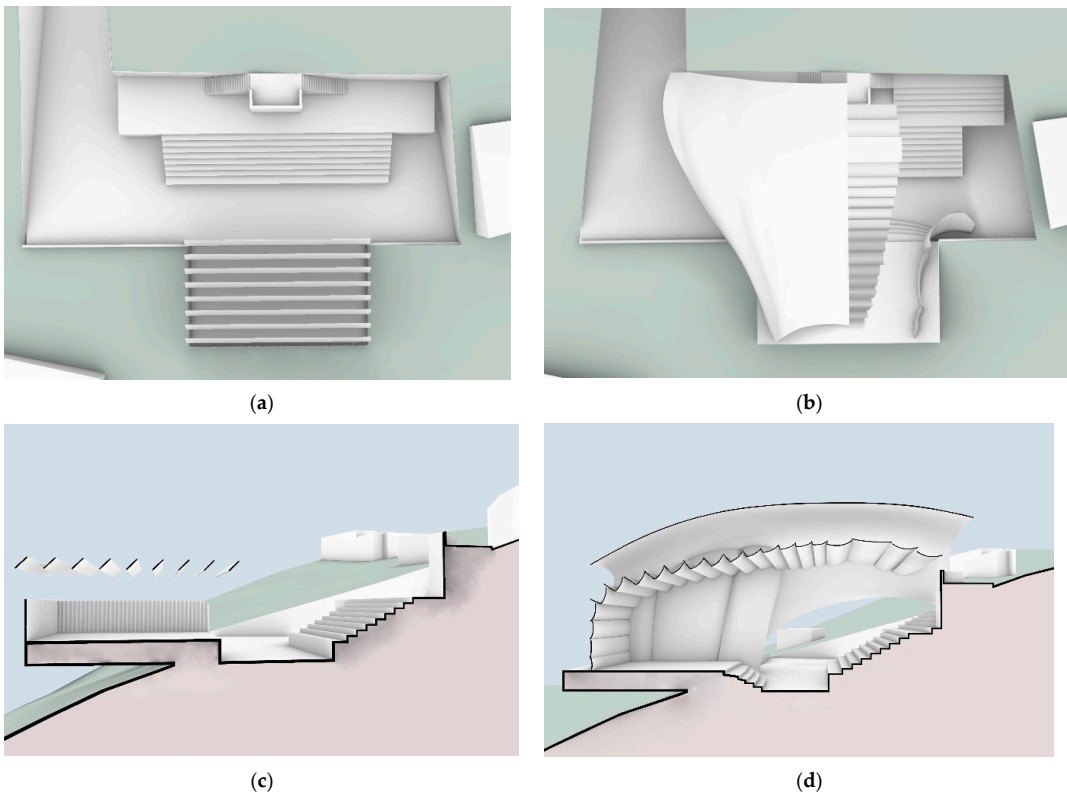

(a)

(b)

(c)

(d)

**Figure 4.** *Cont.*

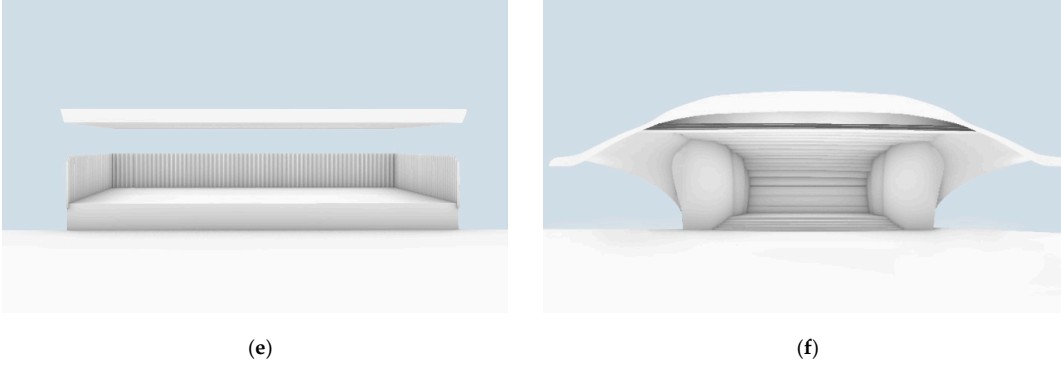

(**e**)  (**f**)

**Figure 4.** 3D model of projects A and B acoustic shell applied to the garden of Rufolo's villa: plan layout of option A shell (**a**), plan layout of option B shell (**b**), transversal section of option A shell (**c**), transversal section of option B shell (**d**), front view of option A shell (**e**), and front view of option B shell (**f**).

The measured results show that the values of the main acoustic parameters with the acrylic panels mounted, both 1.5 m and 3 m high, were not within the optimal range limits for symphonic performances, nor for operas due to the acoustic response being considerably drier.

### 6.2. Option B Shell: Proposal by Authors

Based on the outcomes related to option A shell, the design of a new acoustic shell was proposed to improve the acoustics and make the environment more suitable for opera and ensemble performances. The inspiration for this design came from the nearby sea, which is a predominant element of the natural landscape, constantly under view when on site.

The idea was to create a seashell that ran from the back of the stage to the last row of seats, covering both the performance arts space and the audience area without interruption. Inside the shell, horizontal panels were designed to address the sound energy towards the audience [15]. In addition, the side walls of the stage were designed like petals, by following a convex geometry in order to spread the sound more uniformly, as shown in Figure 4.

Note that the structural stresses were not investigated, which could represent a limitation of this research. The scope of this study was focused on the acoustic response of the room, leaving aside the tension matters, which are very important to complete the design of a new construction but were not the object of this research. Regarding the dimensions, the length of the external envelope covered the extension of the seating wooden rows that were proposed to be built as a second level and integrated with the existing stone staircase used to access the garden where the performance is planned to be.

### 6.3. Materials

As anticipated in the previous section, the stage is currently supported by a scaffolding that levels the inclination given by the hill, as shown in the section of Figure 4c,d. The option A shell is characterized by acrylic for the vertical waved panels and wood for the suspended panels over the stage. The reason for the use of acrylic is that it complies with the intension of transparency, as established by the manager of the festival. The timber used for the suspended panels is due to sustainability since the material is abundantly available on site [16].

The drawing representing the option B shell indicates an airspace between the suspended panels and the external envelope. This is due to the material choice of ethylene tetrafluoroethylene (ETFE) [17], which is lightweight and does not overload the existing scaffolding below the stage floor. This external membrane is flexible and waterproof, under tension at the ends to protect the spectators from sunshine during daylight. It was designed with two arches crossing the longitudinal axis: an arch at the back of the stage and another

wider at the entrance staircase. One of the inconveniences of ETFE membranes is the rain noise that occurs when the water drops hit the smooth and tensed surface of the external skin. Although the shows are programmed only during the summer seasons, a fiber-meshed structure was designed to cover the envelope and mitigate the rain noise in case of adverse meteorological conditions.

The suspended reflecting panels of the option B shell were designed to be composed of recycled pressed paper, a material characterized by high stiffness. This lightweight material was selected to avoid an overload for the existing scaffolding. The acoustic properties of this material have been assumed by considering an experimental application in previous research, since no data are available. The materiality assigned to the digital model has been represented in both absorption and scattering coefficients, drawn as 3D faces, considered opaque on both faces.

## 7. Acoustic Simulations

Before simulating the acoustic parameters with the proposed projects of the new shells, the model calibration was carried out based on the existing condition [18]. The acoustic parameter taken for the model calibration was the reverberation time since the others are more sensitive to the position taken inside the seating area.

The calibration consisted of a loop process that was repeated until the difference between measured and simulated values was within 5% across all the spectrum bandwidths [19]. The negligible difference was due to physical factors that affected the measurements during the survey, like the wind direction and the temperature variance [10]. This was considered acceptable and common for the practice.

The software chosen for the acoustic simulations was Ramsete 3.13, which is based on pyramidal ray-tracing diffusion and supports very well the high level of reflections. Other commercial software available on the market are based on image-source principle. The difference between Ramsete and other software is the time employed for computation that, in general, is dependent on the number of the virtual microphones inserted in the model [20]; however, Ramsete can manage up to 4000 microphones with constant calculation time and can increase the resolution of the acoustic maps based on high level of microphones.

Acoustic simulations were carried out on the digital models with three specific scenarios in order to highlight the contribution and the importance, under an acoustic perspective, of a new shell proposed by the authors (option B) over the panels currently in use (option A) that are temporarily installed until a design of a definitive shell will be in place. All the three scenarios are summarized as follows:

- **Project A**: existing conditions.
- **Project B**: existing conditions with the option A shell. The addition of acrylic panels were floor-mounted at the perimeter of the stage while eight timber panels were suspended above the stage. These timber panels were flat, inclined towards the audience, and installed perpendicularly to the main axis of the space.
- **Project C**: existing conditions with the option B shell. The addition of the new acoustic shell proposed by the authors was composed by an outer layer of ETFE skin, that was covering two sets of diffusing panels: the vertical as side walls, and the suspended ones above the stage and stalls.

All three projects, as shown in Figure 5, were simulated at different percentage of occupancy: 0% and 100% capacity [21]. The characteristics of the three digital models are summarized in Table 1.

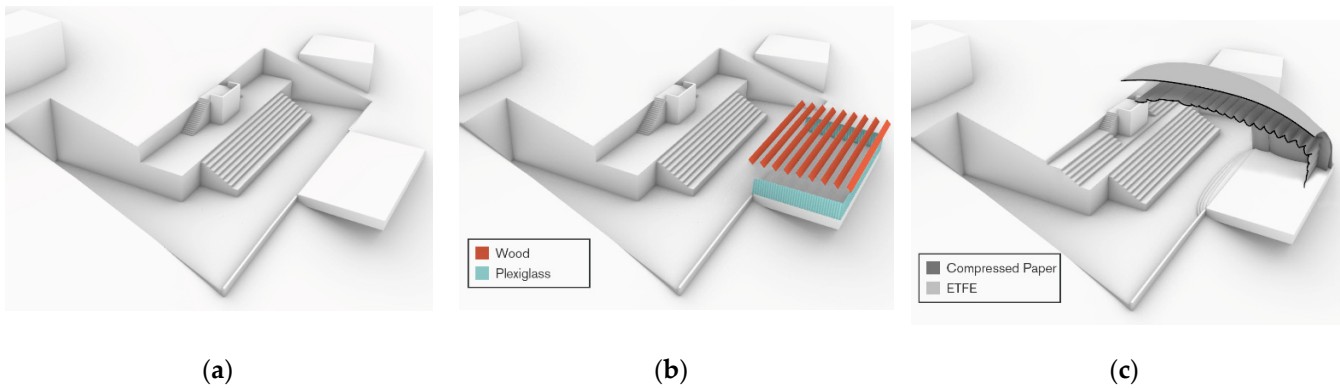

(**a**)          (**b**)          (**c**)

**Figure 5.** Digital reconstruction of the three projects: project A (**a**), project B (**b**), and project C (**c**).

**Table 1.** Specifics of the digital models used for acoustic simulations: existing conditions (project A), addition of temporary panels (project B), and the installation of the new acoustic shell (project C).

| Description | Project A | Project B | Project C |
|---|---|---|---|
| Total number of surfaces | 1386 | 1927 | 1916 |
| Total surface area (m$^2$) | 3368 | 3733 | 6622 |

In total, 192 virtual receivers were distributed over the space, while the sound source was placed in the center of the stage. Table 2 shows the absorption and scattering coefficients of the materials considered in the acoustic simulations.

**Table 2.** Absorption and scattering coefficients of the materials selected for the new acoustic shell.

| Materials | Octave Frequency Band (Hz) | | | | | | Scattering |
|---|---|---|---|---|---|---|---|
| | **125** | **250** | **500** | **1000** | **2000** | **4000** | **(@500–1000 Hz)** |
| ETFE [17] | 0.95 | 0.75 | 0.50 | 0.30 | 0.05 | 0.01 | 0.05 |
| Compressed Paper [22] | 0.15 | 0.10 | 0.10 | 0.14 | 0.18 | 0.24 | 0.15 |
| Solid Timber [16] | 0.10 | 0.10 | 0.10 | 0.09 | 0.10 | 0.12 | 0.05 |
| Tuff stone [16] | 0.01 | 0.06 | 0.05 | 0.02 | 0.08 | 0.05 | 0.11 |
| Terrain [16] | 0.06 | 0.20 | 0.32 | 0.55 | 0.60 | 0.55 | 0.11 |
| Acrylic | 0.10 | 0.04 | 0.03 | 0.02 | 0.02 | 0.02 | 0.05 |
| Audience | 0.51 | 0.64 | 0.75 | 0.80 | 0.82 | 0.83 | 0.37 |

## 8. Analysis of Results

The optimal range of the main acoustic parameters was considered as a function of classical music as the artistic performances represent the main function of the space, which means that the musical response is the subject of the evaluation. The main acoustic parameters were defined based on the standard ISO 3382-1 [8], namely early decay time (EDT), reverberation time (T20), clarity index (C80), definition (D50), and strength (G). These acoustic parameters are presented in the form of diagrams, as shown in Figure 6, and, where appropriate, with acoustic maps as indicated in Figures 7 and 8, which refer to the acoustic parameters that depend on the position assumed in the seating area. The analysis was assessed in the frequency range comprised between 125 Hz and 4 kHz, upon which all the scenarios were analyzed. The results were averaged for all the receivers placed in the digital models.

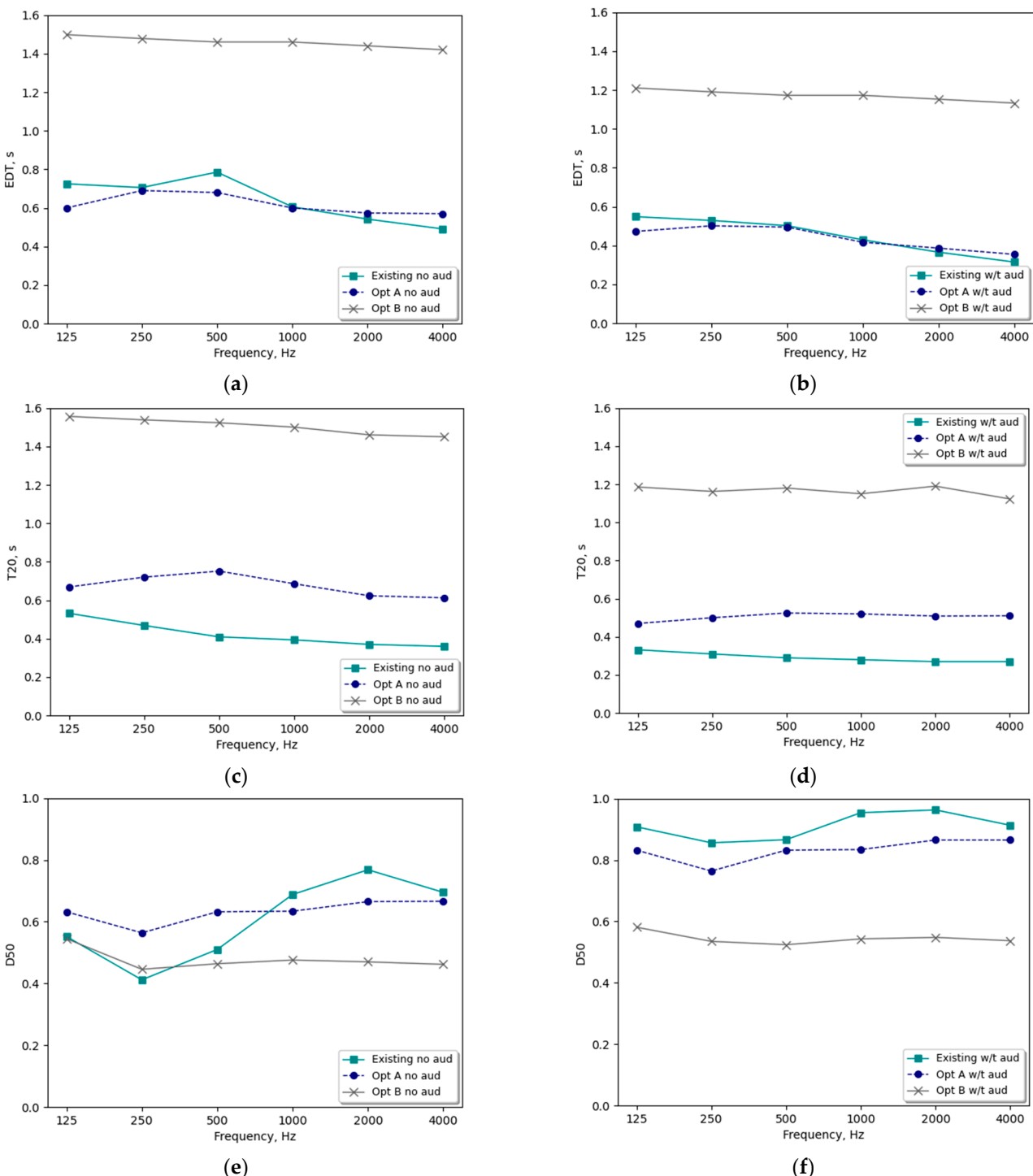

**Figure 6.** Comparison between projects A to C applied to the garden of Rufolo's villa in relation to the acoustic parameters: EDT without audience (**a**), EDT with audience (**b**), T20 without audience (**c**), T20 with audience (**d**), D50 without audience (**e**), and D50 with audience (**f**).

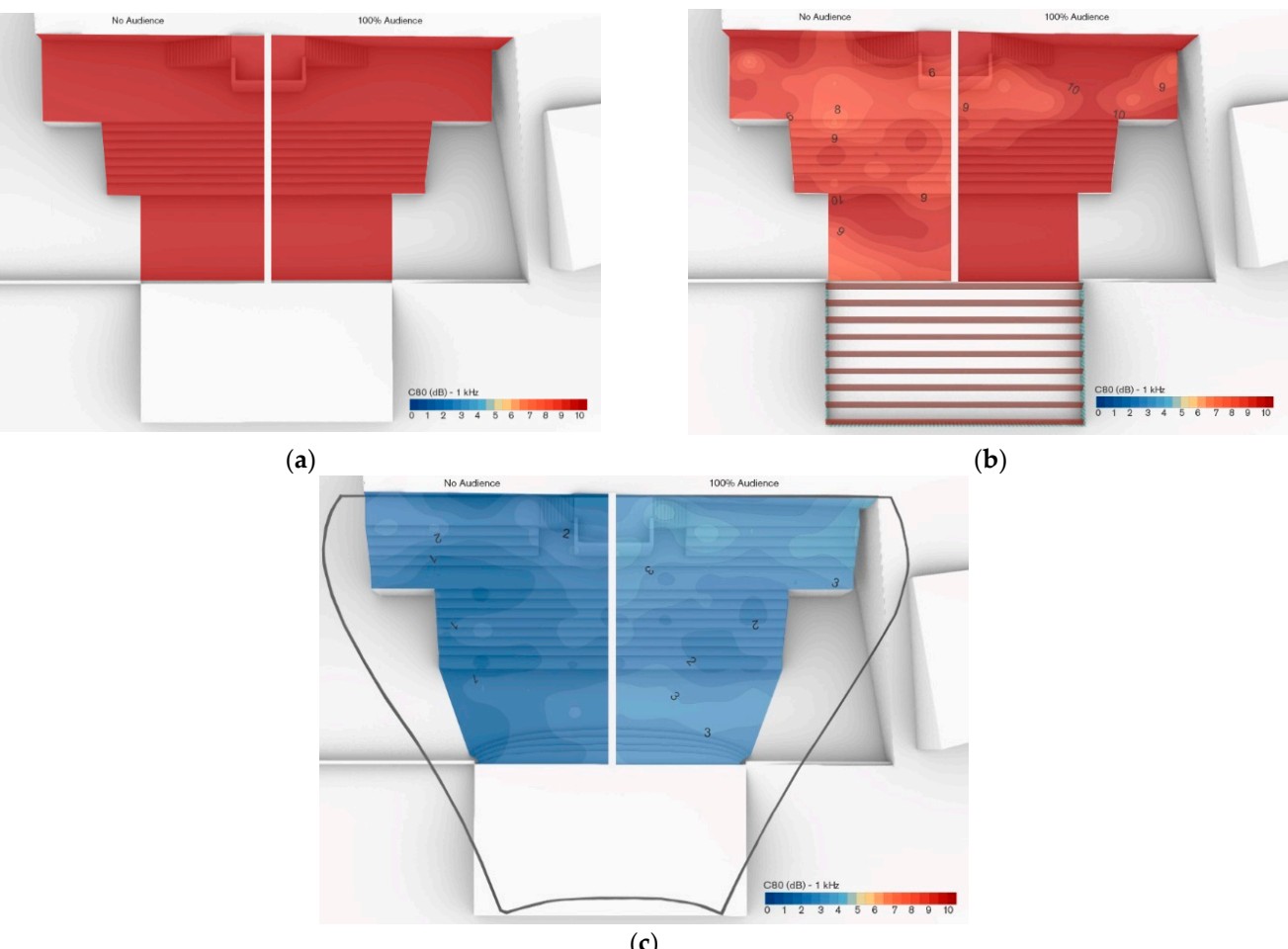

**Figure 7.** Spatial distribution of music clarity index (C80) at 1 kHz, with and without audience: project A (**a**), project B (**b**), and project C (**c**).

Figure 6a shows that the simulated results of the EDT range around 0.6 s, with the exception of the project C with the new shell that is around 1.45 s. A small difference is visible between the existing condition and option A shell, meaning that the addition of the perimeter panels and the suspended ones above the stage do not contribute significantly to the acoustic response of the room. It is visible how the addition of a continuous ceiling that covers all the seating area makes the acoustic response closer to the target indicated by Jordan [11]. With a full audience, the values were attenuated approximately by 0.15 s for the option B shell, as indicated in Figure 6b, but the line trend was similar. The presence of an audience does not make so much difference for the existing conditions and the project A shell, as studied in previous research [4].

Figure 6c shows that the simulated results of T20 are different for the three projects. In particular, the existing condition was around 0.4 s, which is not suitable for musical performance. With option A shell, the T20 values rose up to 0.7 s, which is not yet within the optimal values. The T20 results had a good response for musical performance with the option B shell, which was very uniform across the spectrum bandwidth.

Figure 6d shows that the significant attenuation with the presence of audience is highlighted with the option B shell, to be up to 0.2 s, while the attenuation given by the audience in the other two projects is less effective. In terms of definition, the response in unoccupied conditions, as shown in Figure 6e, ranged between 0.4 (40%) and 0.7 (70%), which means that the acoustic response is suitable for music performance. With an audience in the stalls, Figure 6f shows that only the project with the option B shell remains suitable for musical performance, while the other two scenarios become suitable for speech and

prose, lacking definition for musical shows. As previously anticipated, the clarity index depends on the sound energy arriving within 80 ms (for music) and the reflections arriving in the following decay instants. The ratio determines that the optimal value for clarity is 0 dB, with some tolerance that can range from –2 to +2 dB [23]. Since clarity depends on the location occupied in the audience, the spatial distribution maps are the best representation, as shown in Figure 7, that are related to 1 kHz.

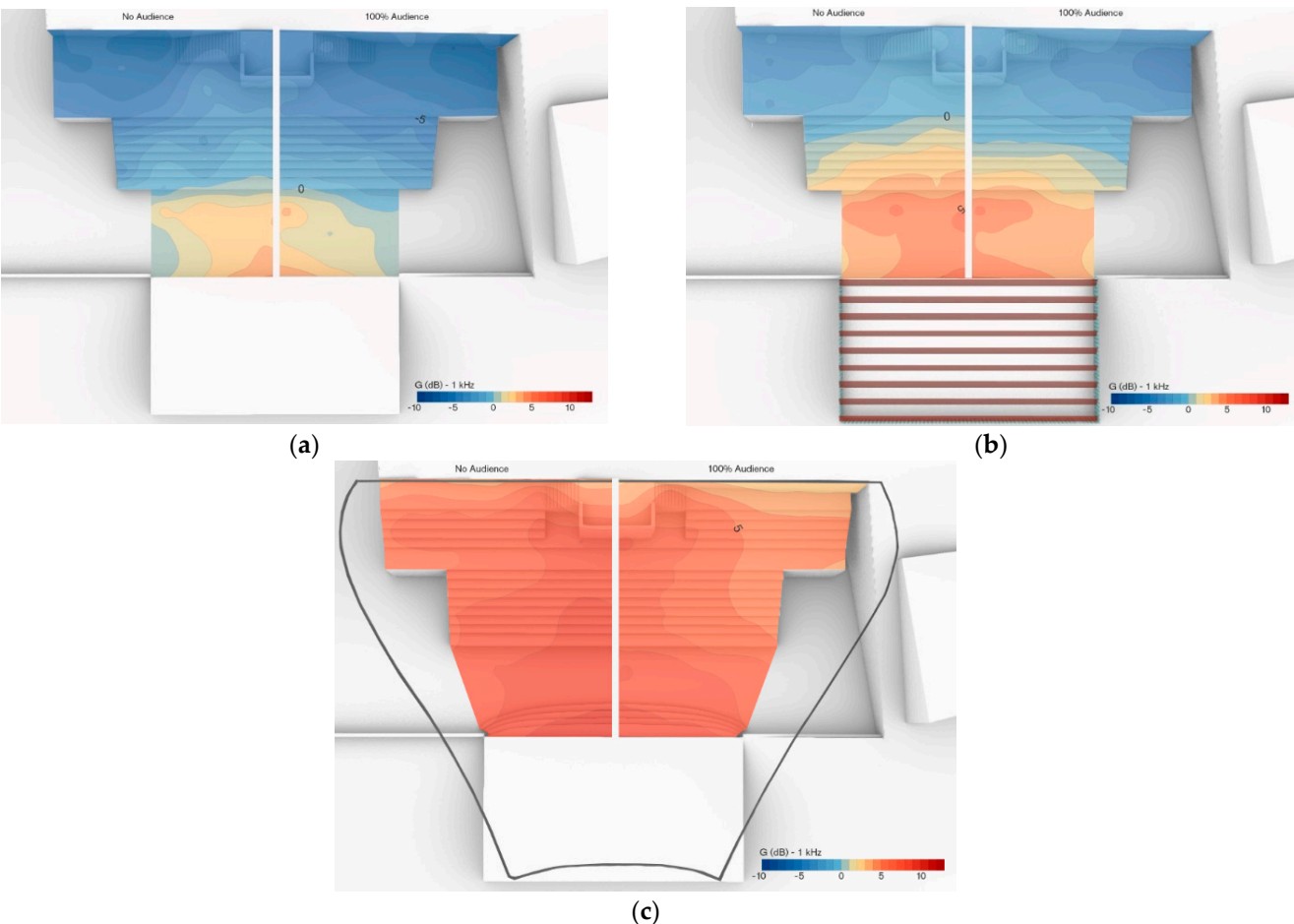

**Figure 8.** Spatial distribution of strength (G) at 1 kHz, with and without audience: project A (**a**), project B (**b**), and project C (**c**).

The graphs in Figure 7 show the plots of the simulated results related to music clarity. In particular, Figure 7a,b show that the C80 values are highly above the upper range limit set by criteria, to be more than 10 dB with scenario A and around 9–10 dB with the type B shell, with no particular difference between 0% and 100% room occupation. With the installation of the type B shell, related to scenario C, the C80 values were significantly lowered and brought to the optimal range, to be between 0 dB and 3 dB in unoccupied conditions and around 2–3 dB when fully occupied, as shown in Figure 7c.

Similar to clarity, the strength (G) also depended on the position of the listener in the seating area [24]. For outdoor spaces, G can have also negative values (below zero) since per definition G equals 0 dB at 10 m from the sound source in free field conditions. For indoors, G can be between 5 dB to 10 dB, mainly based on the interior design and the volume of the room. The graphs summarized in Figure 8 show the simulated results at 1 kHz related to all the considered scenarios.

The simulated results related to strength are shown in the graphs of Figure 8. Figure 8a shows that G values are equal to 0 dB at the first row of seats, and to assume negative values up to the last row, meaning that the reflections given by the only floor are not sufficient to

make the space suitable for good listening conditions [25]. A slightly improved situation was found with scenario B, with the addition of the acrylic panels at the perimeter of the stage and the suspended ones above the stage. Figure 8b shows that the G values range between 4 dB, closer to the stage, and 0 dB found at the last rows of seats. If the extension of stalls on a second level would be in place, this scenario is not sufficient to provide a good listening condition. Figure 8c shows that the G values with the new acoustic shell is found between 10 dB at the front line of the stage to 4 dB at the last rows of seats. If it is considered that the new acoustic shell is designed with the sides open for access reasons, as well as at the back over the staircase, the overall result is very good for a semi-enclosed space, in line with the expectations of this paper.

## 9. Discussion

Overall, the simulated results that compare the current acoustic response with a type A acoustic shell show how poor the acoustics are if no mitigations are in place. In particular, when the side panels are added at the perimeter of the stage in combination with the suspended diffuser panels, the improvement is marginal [26]. Project C shows that the addition of a new shell properly designed by following the acoustic criteria improves the room response to be close to the optimal values set for enclosed performance arts spaces. The lack of any vertical structure behind the stage, a structure very important for directing the sound energy towards the audience, and any ceiling panels of the current state determine the non-suitability of the garden space for live musical performance [27]. These inconveniences determine a reverberation of about 0.4 s, considered averaged across the frequency bandwidth, and a strength to be $-3$ dB, which creates a dry sonic feeling. The installation of the option B shell, as proposed by the managers of the festival on site, determines a very marginal benefit for the overall acoustics, whereas a new acoustic shell, as properly designed under an acoustic perspective, demonstrates the suitability of this space to live musical shows [28,29]. The inclination of the suspended panels at the ceiling, covering both stage and seating area, with the convex side-wall panels, result in a good combination to bring the values of the main acoustic parameters within the optimal range [30–32].

## 10. Conclusions

The design of an acoustic shell is always a result of collaboration between architectural and acoustic studies. The acoustic analysis of the garden within the Rufolo's villa attempts to give a solution to the managers of the Ravello Festival that host important artists at an international level every year. In order to make the space suitable for musical performances, this study highlights the outcomes of the acoustic simulations compared between three specific scenarios: the existing conditions, the addition of acrylic floor-mounted panels at the perimeter of stage in combination with wooden suspended panels, and the installation of a new acoustic shell as proposed by the authors. This study is useful for different applications, explained as follows.

The data obtained from simulations can be employed for auralization as a baseline to create the convolution between IRs and any type of signal that is intended to be auralized potentially in every point of the seats [33]. This can be applied to any type of music performed on the stage, whether the singer or musician is a soloist.

The data of the main acoustic parameters can be taken as a reference for any architectural idea that is addressed to design a different shell and is suitable for the targeted type of performance art [34].

The spatial distribution maps can be considered a preliminary study for the design of an amplified audio system that is going to be installed in the garden of the Rufolo's villa to have as much uniform response as possible.

**Author Contributions:** Conceptualization, A.B., A.T. and G.I.; methodology, A.B.; software, A.B.; validation, G.A. and G.I.; formal analysis, A.B., G.I. and G.A.; investigation, A.B., A.T., G.I. and G.A.;

resources, A.T., G.I. and G.A.; data curation, A.B.; writing original draft preparation, A.B., G.I. and G.A.; visualization, A.B. All authors have read and agreed to the published version of the manuscript.

**Funding:** This research received no external funding.

**Institutional Review Board Statement:** Not applicable.

**Informed Consent Statement:** Not applicable.

**Data Availability Statement:** The original contributions presented in the study are included in the article.

**Conflicts of Interest:** Author Giovanni Amadasi was employed by the company SCS-ControlSys_Vibro-Acoustic. The remaining authors declare that the research was conducted in the absence of any commercial or financial relationships that could be construed as a potential conflict of interest.

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
