# Peer review of "Acoustic Simulations Applied to the Garden of Rufolo’s Villa in Ravello: Comparison between Different Scenarios"

_applsci, doi:10.3390/app14031223_

Round 1

Reviewer 1 Report

Comments and Suggestions for Authors

This is an interesting study of which scope fits within the remit of the journal. I suggest the authors to consider the following points.

Abstract

In the abstract, the authors may wish to give an overview of the study presented, including the research problem/gap, describing the methods used and highlighting the findings, and clarifying why is this study useful and to whom. For example, the description “After a campaign of acoustic measurements, acoustic simulations have been carried out with three specific scenarios, highlighting the effectiveness of the new acoustic shell (project C) over the existing conditions (project A) and option A shell (project B).” is very clear, while the last sentence “Some suggestions have been addressed for how to employ the outcomes of this research study” is too generic.

Introduction

Consider adding a diagrammatic plan to explain the site and its characteristics. This would help readers not familiar with the area to visualise key elements. Option A is mentioned in the abstract, then again in the main text (page 2, line 53) for the first time. It would help to include a brief description of the options A, B and C in the introduction (as they are not explained until Section 6).

The authors may wish to consider adding a brief section in the Introduction to explain the structure of the article, the salient points of the study. (perhaps you can include the three options here)  

Simulation

The method of comparing different design solutions through simulation and measurement is clear and certainly appropriate for the study. However, the manuscript would benefit from more details on the simulation. It is not clear, in fact, how the simulation has been carried out, what type of simulation, what software, parameters, validation criteria, calibration etc. It would be helpful for the authors to explain why they have chosen a specific simulation method over the others and contextualise the choice within a brief discussion over simulation methods (why is the chosen method successful in the context of this study?).

The results of the simulation (Section 8) are clear and well-explained, with clear quantitative indicators to support the findings and the final discussion. I am assuming that authors have used digital simulations. In this case, what algorithms have they used and why?  

Discussion

Consider summarising the findings in the form of bullet point list or a table, to aid the discussion. On page 12, lone 325 “The inclination of the suspended panels at the ceiling […] result a good combination to bring the values of the main acoustic parameters within the optimal range” try to support this point in quantitative terms (the solution is better of XXX% than the other solutions considered etc.).

Conclusions

On Page 10, line 340 “The simulated data can be employed for auralization as a baseline to create the convolution of the impulse response, gathered from the simulations, with any sound signal that is intended to be auralised potentially in every point of the seating area”, consider clarifying the cases to which this solution applies, for example, type of venue, size of the event, type of performance etc.). This would help designers to understand when to follow the design advice indicated in this study.

Figure 2 is very helpful and clear, so are the other figures. 

Comments on the Quality of English Language

The manuscript is well-written and clear. There are, however, very small parts that need some review. Such parts could be checked at the proofreading stage shall the manuscript goes ahead for publication. (examples include page 1, line 12: “which has already been the studied to install some acoustic panels”; page 3, line 85-86: “A campaign of acoustic measurements has been carried out (past tense) in the garden of Rufolo’s villa. The equipment consists (present tense) of an omnidirectional sound source […]”; page 5, line 139 consider changing “plexiglass” with “Perspex” or “acrylic” if relevant etc.).

Author Response

This is an interesting study of which scope fits within the remit of the journal. I suggest the authors to consider the following points.

Abstract

In the abstract, the authors may wish to give an overview of the study presented, including the research problem/gap, describing the methods used and highlighting the findings, and clarifying why is this study useful and to whom. For example, the description “After a campaign of acoustic measurements, acoustic simulations have been carried out with three specific scenarios, highlighting the effectiveness of the new acoustic shell (project C) over the existing conditions (project A) and option A shell (project B).” is very clear, while the last sentence “Some suggestions have been addressed for how to employ the outcomes of this research study” is too generic.7

Dear reviewer, the abstract has been improved.

Introduction

Consider adding a diagrammatic plan to explain the site and its characteristics. This would help readers not familiar with the area to visualise key elements. Option A is mentioned in the abstract, then again in the main text (page 2, line 53) for the first time. It would help to include a brief description of the options A, B and C in the introduction (as they are not explained until Section 6).

Dear reviewer, the introduction has been implemented.

The authors may wish to consider adding a brief section in the Introduction to explain the structure of the article, the salient points of the study. (perhaps you can include the three options here)  

Dear reviewer, this has been provided.

Simulation

The method of comparing different design solutions through simulation and measurement is clear and certainly appropriate for the study. However, the manuscript would benefit from more details on the simulation. It is not clear, in fact, how the simulation has been carried out, what type of simulation, what software, parameters, validation criteria, calibration etc. It would be helpful for the authors to explain why they have chosen a specific simulation method over the others and contextualise the choice within a brief discussion over simulation methods (why is the chosen method successful in the context of this study?).

Dear reviewer, this has been provided.

The results of the simulation (Section 8) are clear and well-explained, with clear quantitative indicators to support the findings and the final discussion. I am assuming that authors have used digital simulations. In this case, what algorithms have they used and why?  

The algorithm is based on image-source, which is added in the pape

Discussion

Consider summarising the findings in the form of bullet point list or a table, to aid the discussion. On page 12, lone 325 “The inclination of the suspended panels at the ceiling […] result a good combination to bring the values of the main acoustic parameters within the optimal range” try to support this point in quantitative terms (the solution is better of XXX% than the other solutions considered etc.).

Dear reviewer, it is difficult to quantify the results with percentage since the results are variable for each octave and are highly dependent on the criteria, which is different for every acoustic parameters. The choice of being discursive is preferred, as common practice. Hoping you can understand, Thank you

Conclusions

On Page 10, line 340 “The simulated data can be employed for auralization as a baseline to create the convolution of the impulse response, gathered from the simulations, with any sound signal that is intended to be auralised potentially in every point of the seating area”, consider clarifying the cases to which this solution applies, for example, type of venue, size of the event, type of performance etc.). This would help designers to understand when to follow the design advice indicated in this study.

Dear reviewer, this has been addressed.

Figure 2 is very helpful and clear, so are the other figures. 

Thank you so much

The manuscript is well-written and clear. There are, however, very small parts that need some review. Such parts could be checked at the proofreading stage shall the manuscript goes ahead for publication. (examples include page 1, line 12: “which has already been the studied to install some acoustic panels”; page 3, line 85-86: “A campaign of acoustic measurements has been carried out (past tense) in the garden of Rufolo’s villa. The equipment consists (present tense) of an omnidirectional sound source […]”; page 5, line 139 consider changing “plexiglass” with “Perspex” or “acrylic” if relevant etc.).

Thank you so much, this has been revised

Reviewer 2 Report

Comments and Suggestions for Authors

This article presents a study of an open concert hall behavior sited in an ancient Italian villa. The paper begins with a brief historical overview, followed by a two-pronged approach to analysis. Firstly, measurements are taken according to ISO 3382 which is not specifically made for the analysis of open spaces-heritage (this specific absence should be in future an opportunity to propose a fourth part of ISO 3382 specific to Heritage and encompassing partially- or fully-open heritage sites). Secondly, acoustic simulations are performed according to three scenarios, two of them with proposed acoustic shells. The paper is well structured and full of content, although there are some minor issues worth noting.

Major issues

There is none and the article has value in presenting itself with this structure, methodology and scope.

Minor issues

In my humble opinion there are some formal issues that need to be corrected, some of them affecting the content:

1. The title seems too prosaic and does not reflect that there is an analysis of the acoustic behaviour of this open space, including 3382, which motivates the decision making on a proposal for an acoustic shell. This is not a major issue, but the article would be improved if the title reflected the seriousness of the study. If it is short, so much the better.

2. In abstract and introduction there are some "the" that needs to be removed; please check the English translation, which is generally very good except for a very few details of this kind.

3. In points 4 and 5, reference is made to the taking of measurements according to the criteria of ISO 3382. This should be mentioned in the title of these points. The following points are also "acoustic measurements" and comments on these results, but due to simulations. This should be made clear in the titles and order of the paragraphs. In this respect, it may be appropriate to place the current point 5 (lines from 102 to 127) under the current point 8, as an analysis of results. The results analysis of the 3382 should be seen in parallel with the results analysis of the simulations.

4. Point 6 refers to project A and project B in relation to the two types of acoustic shells proposed. However, in point 7, it names them differently since it includes "existing conditions" as project A and a project C that was previously project B appears. Finally, in table 1, the three scenarios are named with a nomenclature of acronyms EC, TP, NS, whose only explicit explanation is in the description of the table and not in the text of point 7 (nor in point 6). For all these reasons, it is urgent to tidy up these descriptions that cloud the reading.

5. In figure 6, the comparison with the data from the 3382, which have the value of being taken in situ, is missing. Please include a fourth reference in these graphs with those taken previously and shown in figure 3. It is also necessary to regularise the nomenclature of the three scenarios (as I mentioned in the previous paragraph).

6. In figure 6, the comparison with the data from the 3382, which have the value of being taken in situ, is missing. Please include a fourth reference in these graphs with those taken previously and shown in figure 3. It is also necessary to regularise the nomenclature of the three scenarios (as I mentioned in the previous paragraph). It can be a hybrid nomenclature of "project" and "acronym", but the same throughout the text of the article, please.

7. Point 8 is on results analysis (it could include results analysis of the 3382, too, as I mentioned above). And here is also the discussion of these results. Point 9, which is the actual discussion of results, is a small part of this discussion that has already been made in the previous point and in point 5 (3382). It is therefore appropriate to delete the current point 9 and include it in point 8, which would be renamed "analysis of results and discussion".

8. Conclusions can be stronger

Author Response

Minor issues

In my humble opinion there are some formal issues that need to be corrected, some of them affecting the content:

  1. The title seems too prosaic and does not reflect that there is an analysis of the acoustic behaviour of this open space, including 3382, which motivates the decision making on a proposal for an acoustic shell. This is not a major issue, but the article would be improved if the title reflected the seriousness of the study. If it is short, so much the better.

Dear reviewer, the title has been changed accordingly.

  1. In abstract and introduction there are some "the" that needs to be removed; please check the English translation, which is generally very good except for a very few details of this kind.

Dear reviewer, English has been revised.

  1. In points 4 and 5, reference is made to the taking of measurements according to the criteria of ISO 3382. This should be mentioned in the title of these points. The following points are also "acoustic measurements" and comments on these results, but due to simulations. This should be made clear in the titles and order of the paragraphs. In this respect, it may be appropriate to place the current point 5 (lines from 102 to 127) under the current point 8, as an analysis of results. The results analysis of the 3382 should be seen in parallel with the results analysis of the simulations.

Dear reviewer, this has been added

  1. Point 6 refers to project A and project B in relation to the two types of acoustic shells proposed. However, in point 7, it names them differently since it includes "existing conditions" as project A and a project C that was previously project B appears. Finally, in table 1, the three scenarios are named with a nomenclature of acronyms EC, TP, NS, whose only explicit explanation is in the description of the table and not in the text of point 7 (nor in point 6). For all these reasons, it is urgent to tidy up these descriptions that cloud the reading.

That’s true, there was confusion between option and project in identifying each scenario, which now has been tied up.

  1. In figure 6, the comparison with the data from the 3382, which have the value of being taken in situ, is missing. Please include a fourth reference in these graphs with those taken previously and shown in figure 3. It is also necessary to regularise the nomenclature of the three scenarios (as I mentioned in the previous paragraph).

Dear reviewer, once the calibration has been made, it is possible to compare directly the simulated values of the existing conditions with the other 2 scenarios. Like anticipated in the calibration process, the drift between measured and simulated data exists, so it has been evaluated to compare directly all the simulated values.

  1. In figure 6, the comparison with the data from the 3382, which have the value of being taken in situ, is missing. Please include a fourth reference in these graphs with those taken previously and shown in figure 3. It is also necessary to regularise the nomenclature of the three scenarios (as I mentioned in the previous paragraph). It can be a hybrid nomenclature of "project" and "acronym", but the same throughout the text of the article, please.

Dear reviewer, the regularization has been done

  1. Point 8 is on results analysis (it could include results analysis of the 3382, too, as I mentioned above). And here is also the discussion of these results. Point 9, which is the actual discussion of results, is a small part of this discussion that has already been made in the previous point and in point 5 (3382). It is therefore appropriate to delete the current point 9 and include it in point 8, which would be renamed "analysis of results and discussion".

Dear reviewer, the structure of the article was highly appreciated by the reviewer 1, and the authors would like to keep in this way, without interfering with the comments of reviewer 1. Hoping you can understand.

  1. Conclusions can be stronger

Conclusions have been slightly adjusted.